# Identification of Network Promoters in a Regional and Intersectoral Health Promotion Network: A Qualitative Social Network Analysis in Southern Germany

**DOI:** 10.3390/ijerph18168372

**Published:** 2021-08-07

**Authors:** Tobias Fleuren, Ansgar Thiel, Annika Frahsa

**Affiliations:** 1Department of Social and Health Sciences of Sport, Institute of Sport Science, Eberhard Karls Universität Tübingen, 72070 Tübingen, Germany; tobias.fleuren@uni-tuebingen.de (T.F.); ansgar.thiel@uni-tuebingen.de (A.T.); 2Institute of Social and Preventive Medicine, University of Bern, 3012 Bern, Switzerland

**Keywords:** intersectoral cooperation, participation, social network analysis

## Abstract

Health in all policies is a key approach to promote health and calls for cooperation between diverse levels of government and different sectors. In this paper, we analyze how a network called ‘Healthy Region Plus’ in Southern Germany addresses intersectoral cooperation at city and county levels. We aim to analyze the different roles of actors involved in the network based on the promoter model. We conducted two socio-material network mappings based on the Net-map approach by Schiffer and Hauck. The analysis followed three steps: data visualization, descriptive analysis of network properties, and interpretation of findings. Our findings reveal a complex intersectoral cooperation structure, with county and city level clusters, with network members who act as diverse power, expert, process, or relationship promoters. We also identified certain relevant sectors not to be part of the network. We discuss that the success of the network depends on the members’ active participation in and their outreach beyond the existing network, between city and county levels, and across sectors to promote health and build health-promoting structures in the region.

## 1. Introduction

In modern society, health has become a relevant topic in diverse policy sectors from medical care, transport, social affairs, education, and finance to employment and housing [1,2]. These different responsibilities for health in various policy sectors make cooperation necessary between sectors when it comes to health promotion [3,4]. This cooperation in health promotion also needs to consider the different levels of society including the neighborhood, in order to promote health in those settings where people “learn, work, play and love” [5] (p. 4). Within health-in-all-policies approaches, though,—at least social—ownership tends to remain with the health sector, which calls for integration developments at the interface between health and non-health sectors [6].

Despite these manifold integration calls, integration through intersectoral cooperation has proven to represent a challenge in practice [4]. Barriers to intersectoral action have been identified when it comes to lack of political leadership and political will, but also when it comes to finding a common language and defining shared goals for all partners [7]. Furthermore, a functioning cooperation can be hindered by siloed thinking of partners and differences in the objectives and interests among the involved actors of the different policy sectors [8]. Concerning facilitators of intersectoral health promotion, research on the level of local governments has shown that intersectoral network structures represent a main prerequisite of functioning intersectoral cooperation [3]. Intersectoral network structures allow participatory planning (priority setting, joint evaluation) based on shared mandates (protocols, regulations, frameworks) and using shared financial tools [3]. In these networks, actors from different policy sectors and levels of policymaking have to collaborate.

The objective of this exploratory study was to identify different types of network promoters and to understand their specific contributions to fostering network cooperation and, ultimately, to intersectoral health promotion. To do so, we will first link intersectoral cooperation to social network relationships and build upon the promoter model as an approach to social network analysis that allows us to analyze different types of promoters within a given network. We will then introduce a region for health, called ‘Healthy Region Plus’ (Gesundheitsregion^plus^ (GR+)), in the state of Bavaria, Southern Germany, as an empirical case study for intersectoral cooperation in health promotion. Healthy regions strive to improve health promotion and health care by developing strategic alliances and cooperation that foster good governance and implement interventions at regional and local levels. We will use a participatory mixed-methods-approach of social network analysis to investigate the different actors from diverse policy sectors, artifacts created, and roles taken within the network.

In the results section, we will present the specific network structures, as perceived by different perspectives and different types of network promoters and their contribution to intersectoral cooperation. We will close the paper by discussing the results against the theoretical background and conclude with the transferability of findings to other networks.

### Theoretical Background

The quality of cooperation in intersectoral relationships is highly dependent on social network relationships. Social networks describe relationship structures between interacting partners. The relationships can be both formal and informal [9]. In formal organizational networks, professionals are usually connected through vertical (hierarchical) and horizontal (task-specific) relations. These relationships are usually laid down in job and work plans. Beyond these formal relationships, however, informal network relationships often form between the respective job holders, which can have quite a high influence on the fulfilment of joint work tasks [9]. In practice, informal networks sometimes even have a greater influence on work-relevant contexts than formal relationships.

The main function of social network analysis is to make structures and dynamics within existing networks visible [10]. Network analysis has been emerging in public health research and health promotion since the 2000s [11,12] and has applied to various sub-fields such as research on agenda-setting by health policy coalitions in chronic disease prevention [13] creation and implementation of local wellbeing policies [14], collaboration between public and community organizations [15], or middle managers’ roles in inter-agency governance [16].

While quantitative approaches have been particularly prominent in social network analysis, with their main benefits being a reduction and aggregation of complexity regarding social ties and relations [17], there has also been a growing interest in qualitative approaches. Qualitative approaches allow for an insider-perspective and can reveal network dynamics at a micro-level [18]. Over the last years, an integrated approach of qualitative and quantitative elements has been suggested [17] to use the strengths of both approaches.

For our study, the so-called promotor model is particularly helpful for social network analysis because it is suitable to identify network actors who function as drivers for innovations, both on the formal and on the informal level. The promoter model was originally created to explain the dynamics of innovations [19]. In recent years, it has been applied to a variety of social networks and continuously updated by different researchers [20,21,22]. Within the context of health promotion as an integration of diverse actors and sectors, the promoter model serves to understand the specific contributions of different network members to foster complementary and different levels of governance, policy, and action [6].

The promoter model differentiates four types of promoters, according to their functions within network structures. They can be defined by “the type of barriers they help to overcome, the type of power bases on which their influence is grounded, and the type of characteristic value-creating functions they fulfil by their specific type of behavior” [21] (p. 409).

(1) A power promoter is important to overcome the unwillingness of actors to engage in a network or a project [20]. Due to their hierarchical standing within an institution, power promoters are able to provide needed resources and shape policy and action [21]. (2) In contrast, technology or expert promoters distribute professional knowledge and skills. In this function, expert promotors help to overcome the barrier of ignorance by applying specific technical knowledge” [20] (p. 41). (3) Process promoters tackle the “barriers of non-responsibility and indifference which are primarily caused by organizational and administrative resistance” [20] (p. 42). (4) These three types of promoters mainly focus on processes within their organizations. For contacts and cooperation to external partners, relationship promotors are needed [21]. This type of promoter fosters collaborative governance of a network, brings together diverse partners, organizes the exchange between them, and manages arising conflicts in cooperation [22].

The promotor model will be used as the theoretical basis for empirically analyzing the intersectoral network of the Healthy Region Plus with regard to the different roles of actors, their participation, and their relevance for the progress and development of the network.

## 2. Materials and Methods

### 2.1. Study Setting

Starting in 2015, the Bavarian State Ministry of Health and Care has provided funding to strengthen the regional development of structures and networks for health promotion, prevention, and health care [23,24,25,26,27,28,29]. The funding program ‘Healthy Regions Plus’ gave Bavarian cities and counties in Southern Germany the possibility of applying for funding to build new regional alliances for health. The University of Tübingen has been commissioned with the academic support and evaluation of one of the funded Healthy Regions Plus. This specific Healthy Region Plus represents a cooperation between the city and county-level and is based on the principles (1) Equity in health, (2) Intersectoral cooperation, and (3) Participatory collaboration among all actors associated with the Healthy Region Plus [30].

The Healthy Region Plus consists of a project management office that is responsible for steering the program of the region, managing the day-to-day business, and supporting all activities related to the Healthy Region Plus. The two project managers are equally employed by the county and the city and have backgrounds in social work/public administration and physiotherapy/health management.

The steering committee of the Healthy Region Plus is the specific unit of our analysis because it represents a strategic network responsible for intersectoral cooperation through several actions (e.g., broad participation of actors, commonly defined goals) that supervises all content-related and strategic planning.

Currently, the steering committee consists of local politicians (major, city council members, county council representative), employees of the county health department and of the local sports department, the office for socio-cultural affairs of the city, and two volunteer associations in the field of health and medicine (see Table 1). The regular meetings of the steering committee are organized by the project management office.

The thematic priorities of the Healthy Region Plus are categorized within five fields of action: medical supplies, peer mediators, healthy working environments, healthy living environments, and low-threshold professional help for people in difficult life situations. The actual work is organized into corresponding working groups and projects such as a working group for single-parenting issues or a midwifery center.

### 2.2. Data Collection and Analysis

Following an interpretative epistemological approach, we used qualitative network analysis, applying the Net-map [31] method, to study the multiple realities, descriptions, and experiences of diverse relevant actors involved in an intersectoral network for health promotion.

While the majority of studies using qualitative network analysis has focused on the private networks of individuals (e.g., transnational migration–related family networks, biographical networks) [32] few studies so far have used qualitative network analysis in research about institutions or the relationship between individuals within institutions. Among those few that have dealt with institutional actors, Schiffer and Hauck [31] have presented an elaborate method of how to conduct qualitative network analysis: the Net-map method.

The Net-map method was originally developed in international agricultural research as an interview-based mapping tool with multifold purposes, among these are the visualization of implicit knowledge, the understanding of the interplay of complex formal and informal networks, power relations, and actors’ goals, identification of conflicts and potentials for cooperation as well as knowledge exchange and facilitation of learning processes [31]. The method has been widely used in (network) methodological research [33,34,35], socio-ecological and sustainability research [36,37], and health research [38,39,40].

In the Net-map method [31], network members are asked to identify all relevant actors, to link the actors, map their influence, and to discuss the network map created.

To do so, they are asked to name and write down on cards all of the relevant actors and their tasks, their involvement in working groups, and collaborations on artifacts (such as events, products, services, projects) within a network.

Furthermore, the participants are asked to put communication structures, conflicts, and relations to external stakeholders in writing. This process of identifying network structures is moderated and structured by a moderator. Afterward, the participants are asked to use stackable checker pieces such as stones to build influence towers that indicate the influence and the importance of each network member (the more influence an actor has, the higher the tower). In a last step, the final network map is discussed and critically reflected by all participants.

The process helps participants to understand, visualize, monitor, evaluate, and improve contexts in which different actors shape processes and outcomes. The Net-map method thus combines visual results (network maps), quantitative results (network data such as centrality measures), and qualitative results (network narratives). It also understands network members as participants and active agents rather than research objects.

We conducted two separate sessions, one with the project management office and a separate one with the steering committee members to identify overlaps and differences in their perceptions of the members’ contributions and interactions and to use those to facilitate discussion on the different types of support and contributions that might be needed to foster intersectoral cooperation in this network. We also conducted the mapping with project office managers separately from the other steering committee members to allow them to express their perceptions and assessment of influence, being aware of potential power imbalances as project office managers were in an (indirect) work-related dependency from certain steering committee members.

We conducted a first network-mapping with the two project office managers (Geschäftsstellenleitung). The third participant of the mapping (expert in health promotion in the county), was employed at the regional health department and is a representative for regional health promotion in the network. This member had also acted as a temporary substitution in the administration office during the leave of a manager.

With the steering committee members (cf. Table 1) present in a regular meeting of the steering committee, we conducted another mapping. In this mapping process, each participant indicated their interactions with other stakeholders (internal and external) and their own contributions to the ‘Healthy Region Plus’ program. Both network mappings were group-based interviews.

We conducted the study according to the guidelines of the Declaration of Helsinki, and approved by the Ethics Committee of the Eberhard Karls Universität Tübingen (protocol code AZ:A2.5.4-075_aa, 6 June 2018). All participants provided written informed consent for participation.

Data analysis. The data analysis followed two steps [12]: (1) Visualization of the network and (2) description of the network properties. For the first step, we used the open-source software gephi [41] to transfer hand-written maps into computer-generated maps. Using gephi, it was possible not only to visualize the network, but also to calculate the centrality measures.

We calculated three types of metrics: the degree, the density, and the betweenness. The degree centrality informs about the centrality of an actor by indicating the number of direct connections to him [12]. ‘Density’ measures the number of existing ties in relation to the number of all possible ties [42]. The betweenness centrality indicates whether other actors are dependent on an actor [43], respectively, the “extent to which an actor lies between two nodes that would not otherwise be connected” [12] (p. 74).

In a third step, the statistic results were contextualized by findings of qualitative interviews that had been carried out with the steering committee and the project management office parallel to the network mappings. Interviews were transcribed verbatim and analyzed thematically, supported by MAXQDA software (VERBI Software GmbH, Berlin, Germany).

## 3. Results

The specific form of cooperation between the participating actors in the ‘Healthy Region Plus’ represents a collaboration network that is characterized by the “willingness to work together and […] intensive contacts and communications between the different organizations” [44] (p. 80). In the following, we will present findings on the network structure, as perceived by the project office managers and by the members of the steering committee. We will then present the different promoter roles identified in the ‘Healthy Region Plus’ network and how these different promoter roles might explain the differences in perceptions by the project management office and the steering committee members. The promotor roles will be derived according to the theory [20,21]. The official status of an actor within their organization contributes to the classification of an actor.

### 3.1. Network Structures from the Perspective of the Project Office Managers

The network map created by the project office managers consists of 31 nodes (actors/content) and 54 edges (relationships between actors) (Figure 1).

Segregation and homogeneity. The assignment of membership of relevant actors to networks of city and county shows a high degree of segregation (Figure 2). From the perspective of the project management office, the actors in the city are sector-related, more diverse, and better connected to each other. While the county network mainly consists of politicians and members of the regional health department (the place of employment of the office manager), the network of municipality-members is characterized by a higher degree of differentiation (e.g., politicians, NGOs) and is related to more heterogeneous working fields. Regarding the criterion betweenness, only the office managers, the chairperson of the senior advisory board of the city, and the representative for the regional health promotion in the county are involved in networks that extend beyond their own authority.

Degree and betweenness. As to be expected, the office managers stated that they played a crucial role in the communication of the network. An evaluation of their degree and betweenness measures validated the first impressions that the network map presented. The two office managers were not only assigned the most direct contacts (office manager A = 12; office manager B = 10 contacts; see Table 2), but also the highest betweenness-rate. This can be explained by the fact that they are the initiators and facilitators of the meetings.

From the perspective of the project office managers, some members of the steering committee (a researcher, three politicians, a chairperson of a local volunteer health association, a chairperson of the senior advisory board and the director of the municipal sports department) function as representatives for health promotion in their original work contexts. In this regard, they explicitly take over the networking task. The particular importance of these actors is mirrored in their number of direct network connections (five to six each) and their betweenness rates. One of the politicians, who had a comparably lower rate of direct contacts, can be considered to have the function as a bridge-builder to other actors. The chairperson of the senior advisory board is an important contact to representatives. Politicians A and D were both important with regard to the structural development of the region of health, however, politician D was also engaged in contributing with content-related activities.

### 3.2. Network Structures, as Perceived from the Perspective of the Steering Committee Members

The network map by members of the steering committee consisted of 47 nodes and 59 edges. Compared to the map by members from the project management office, the main differences referred to centrality measures within the network.

Degree and Betweenness. From the perception of the steering committee members, the role of the project office managers was different from the one perceived and visualized by the project office managers in their own mapping. The main differences refer to the degree and betweenness centrality, which were significantly lower compared to the mappings of the project office managers (see Table 2). Only project office manager A was assigned a comparatively high level of degree and betweenness centrality in both maps, the one created by the steering committee members and the one by the project office managers themselves. In contrast to the perception of the project management office, the steering committee assessed the number of direct contacts by office manager B only at two, and in comparison, the importance of the betweenness-centrality was also less important in the mapping.

Furthermore, project office managers were not recognized as a main linkage between municipality and county, which had been the case in their self-mapping.

The actors who were assigned the highest centrality measures where the two politicians and the chairperson of the senior advisory board, who were considered as holding multiple contacts, especially to other politicians, diverse volunteer associations, administrational structures, and general practitioners (Figure 3).

Potentially relevant sectors for health promotion were missing or underrepresented sectors in the network such as housing, mobility, environment, and urban planning. In addition to non-participation, the network mapping also highlighted that network members did not outreach beyond the network members to those sectors, at least not related to the ‘Healthy Region Plus’.

### 3.3. Promoter Roles within the Steering Committee

The analysis of both mappings from the perspective of the promoter model revealed that the main four different promoter roles were present in the ‘Healthy Region Plus’.

Power-promoters. There are several network members who showed particularly high degrees and betweenness centrality, as perceived by the steering committee members (cf. Table 3). These might be labelled as power-promoters: the chairpersons and the directors of the municipal sports and the county health departments have the power to distribute the goals and strategies of the intersectoral health region network within their respective organization, while politicians A and D can have a direct impact on decisions within the county and the municipal administrations due to their leading roles in these organizations.

Technology promoters. The office managers and the representative for health promotion in the county also showed high degrees and betweenness centrality, as measured from their perspective. Their centrality is linked to a promoter function based on their task descriptions as office managers, that is, to spread the basic principles of the Healthy Region Plus (e.g., intersectorality and participation) and in this sense, give advice for the development of strategies for health promotion within the municipality and the county.

Process promoters. The office managers can also be considered as process promoters because they are responsible for the administration of the whole Healthy Region Plus project and therefore have the potential to overcome bureaucratic and institutional barriers by going through unofficial channels and in this sense, ‘cut through the red tape’. This also holds true for the supporting office member.

Relationship promoters. Office manager A was the only network member who showed high degrees and betweenness centrality, as measured from the perspective of the steering committee members as well as from the project management office. The office manager A was perceived by the steering committee members as a mediator between the diverse interests of the network partners and as the one building relationships to recruit cooperation partners and promote these relationships in the interest of the Healthy Region Plus.

## 4. Discussion

In this study, we examined intersectoral cooperation in a Healthy Region Plus network in Southern Germany by using a participatory mixed-methods network mapping approach. Previous studies on intersectoral action have often concentrated on the factors that may influence cooperation in a network (e.g., organizational aspects or leadership) [3,7,45,46,47,48,49,50,51]. Several studies have analyzed networks (formalized and non-formalized) with regard to their structure and statistical measurements, without pointing out the roles and influences of certain actors, like we did in this study [52,53,54,55]. Other studies have focused on the evaluation of community partnerships and corresponding health outcomes and impacts [46,56,57]. However, little is known about the contributions of certain actors in public health networks. The literature agrees on the importance of leadership; Roussos and Fawcett [48] pointed out that in external funded projects, this role is mostly tied to the manager of the project, usually an employee of the project. This is also true for the network structures studied in this paper, in which both employed project office managers shared responsibility for the project and the network, which led to a high overall connectivity within the steering committee.

To gain more insights into the structures in the Healthy Region Plus, we analyzed which actors were involved in the steering of this project and which roles they had. Our network analysis showed quite a complex intersectoral cooperation structure with typical sectors represented: the sectors involved in the Healthy Region Plus mirror the ones that other studies have identified such as the one by Rantala and colleagues [3]. They identified the governmental sports and social affairs sector, community representatives, and NGOs as the most involved actors in local governmental structures in the area of health promotion interventions [3]. As with the sectors participating, our study confirmed some sectors as potentially relevant but missing or underrepresented sectors in the network, which has also been the case in other studies on similar topics [8].

With regard to the promotion of innovation, Hauschildt et al. [20] suggest a “troika” model of a technology, a process, and a power promoter as the most promising combination for innovations in intersectoral networks. In our study, the actors most heavily involved in steering the intersectoral networks acted as power promoters by exerting influence on higher intraorganizational-levels. We also found that the most relevant promoters (the project office managers) were also highly involved in operative activities and covered a relationship-, a technology- and a process-promoter role at the same time.

Our findings show that the success of the Healthy Region Plus mainly depends on the relatively high engagement of steering committee members, as indicated by the participation of all members of the steering committee in developing a health strategy and defining rules of procedures. At the level of the steering committee, the Healthy Region Plus project was well structured and organized, could rely on the support of two project managers/coordinators, and received support from political decision-makers. These factors have also been listed in the literature as prerequisites for fruitful cooperation [50,58]

Furthermore, in all activities, both the political committees of the city and the county were included in the strategic planning of the Healthy Region Plus. The fact that actors in collaborative network relations often take more than one role has also been observed in other studies [59]. In a study by Wijenberg et al. [59], the overlapping functions even turned out to be facilitators of collaborative learning in network structures [59].

In this sense, it can be stated that promoter roles play a very important role in successful intersectoral collaboration. These roles do not necessarily have to be attributed to different persons. What is important, on the other hand, is that all sectors involved are also actively involved in the process of steering the intersectoral network, starting with the development of a shared vision and ending with a reflection on the possibilities and limits of concrete implementation strategies for health promotion programs.

### Study Limitations/Strengths and Weaknesses

Social network analysis cannot reveal all aspects of collaboration in intersectoral networks. What cannot be answered, for example, is the question of how relevant personal experiences and the motivation of the participating actors are for successful network collaboration at the level of a steering committee. At the same time, it cannot be ruled out that subjective experiences and motivations influence the perception of actions and dynamics within the analyzed network.

Another limitation of our study was the execution of the network mappings. The project management office was interviewed as a group, which made the work process more dynamic, but also impeded the participation of some members. The mapping with the steering committee was also planned and conducted as a group interview, but due to time restrictions, not all members of the committee could participate in the mapping process. Two members of the steering committee missed the network mapping. In addition, the project office managers were only present during the network steering committee mapping and did not participate in this mapping. Given that missing data/nodes can have significant effects on the network structures captured, missing actors and attributes they might have added to the mapping may have produced a different network map. These are challenges that group-based mapping methods bring with them [60]. We tried to overcome these challenges in the mapping by explicitly asking each participant for the contributions and interactions of the other members and by focusing on the network interpretation of identifying different promoter types in the Healthy Region Plus to understand their specific contributions to fostering network cooperation and, ultimately, to intersectoral health promotion.

One strength of our study is the qualitative element of the network mapping approach. The qualitative approach made it possible to gather in-depth information about the Healthy Region Plus. In particular, the explanations given while drawing the networks helped to contextualize the findings of the mappings. Herein, our approach does not aim to gain representational generalizability like quantitative study designs might do, but, in accordance with Smith [61] at achieving transferability as a specific type of generalizability, acknowledging that “knowledge is constructed and subjective, reality is multiple, created and mind-dependent, and methods cannot provide theory-free knowledge” [61] (p. 140). Carminati [62] underlines this understanding of qualitative research. She stated that qualitative approaches aim to gain in-depth information of the meanings and processes in the respondents’ everyday life. Qualitative research in an interpretivist tradition reveals its strengths by “the understanding of how individuals, through their narratives, perceive and experience their lives, constructing meanings within their social and cultural contexts” [62] (p. 2096). Regarding this understanding of qualitative research, the scope was on the transferability of the findings. While the perceptions of the respondents and the findings of the network analysis may be specialized, the design of researching dynamics of intersectoral cooperation is adaptable to other settings.

For future research on intersectoral collaboration, a combination of network mappings with qualitative interviews is recommended. Furthermore, a longitudinal approach could help to grasp changes in network set-ups, the actors involved, and promoter roles, but also changes regarding the influence on outcomes of the network collaboration over time.

## 5. Conclusions

Regional networks such as the Healthy Region Plus presented in this study depend on cooperation between different sectors as well as between the city and county levels of government. Relationship promoters, here, a project office manager, can span those boundaries and contribute to building a common understanding of the Healthy Region Plus as a network to promote health equity at the regional level. Power promoters, here, local politicians, are decisive as drivers for network progress and perception of the network in different political institutions, while process promoters guarantee maintaining activities within the thematic scope of the network.

## Figures and Tables

**Figure 1 ijerph-18-08372-f001:**
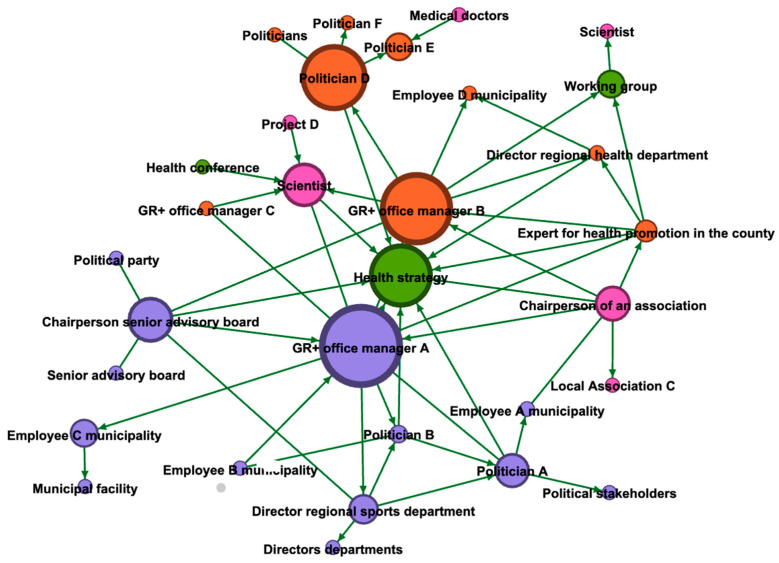
Map from the perspective of the project office managers. The size of the nodes displays the betweenness centrality of the actors. Orange nodes indicate actors of the county, purple nodes indicate actors of the city, pink nodes are related to actors that cannot be categorized into city/county, green nodes indicate artifacts (events, products, services, projects).

**Figure 2 ijerph-18-08372-f002:**
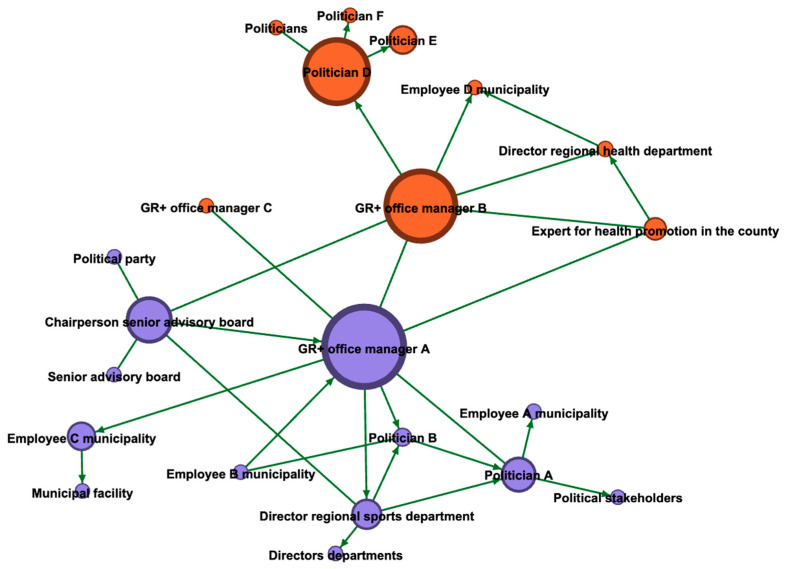
Map from the perspective of the project office managers split by actors from the city (purple nodes) and the county (orange nodes).

**Figure 3 ijerph-18-08372-f003:**
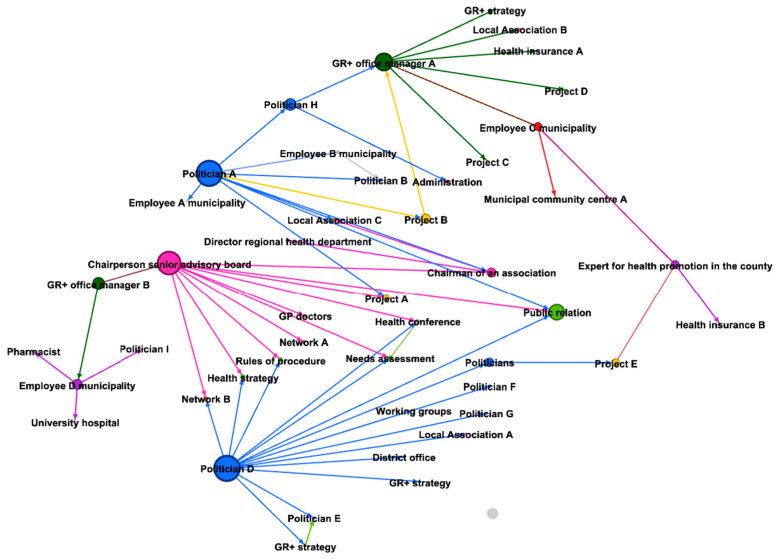
Map from the perspective of the steering committee members. The size of the nodes displays the betweenness centrality of the actors. Blue nodes indicate politicians, yellow nodes indicate projects, pink nodes indicate associations, brown nodes indicate external contacts, purple nodes indicate actors from the public health service, red nodes indicate members of the public administration, nodes in darker green are the members of the project management office of the Healthy Region Plus, the nodes in a lighter green indicate events of the strategy of the Healthy Region Plus and the grey nodes are others.

**Table 1 ijerph-18-08372-t001:** Official members of the steering committee.

Actor	Function	Area, Sector
Office Manager A *	Administration and project management of the ‘Healthy Region Plus’	City, Employee of the sports department
Office Manager B *	Administration and project management of the ‘Healthy Region Plus’	County, Employee of the health department
Expert for health promotion in the county *	Representative and officer for health promotion and health equity	County, Employee of the health department
Director regional health department	Official lead of the ‘Healthy Region Plus’	County, Employee of the health department
Chairperson senior advisory board *	Representative for the elderly and medical topics, General practitioner	City, Volunteer association, Part of the city council
Politician D *	Representative of the county	County, Part of the county council
Politician A *	Representative of the city	City, Major—Part of the city council
Chairperson of an association	Representative for medical issues and health promotion, General practitioner	City/County, Volunteer association
Employee C municipality *	Bridging function to communities and underrepresented population groups	City, Employee of the office for socio-cultural affairs

* Actors of the steering committee who participated in the network mapping.

**Table 2 ijerph-18-08372-t002:** Degrees and betweenness centrality as measured from the perspective of the project office managers; displayed are only actors with a degree > 1; Density = 0.116).

	Actors/Topics	Degree	Betweenness Centrality
1.	GR+ office manager A	12	144.57
2.	GR+ office manager B	10	121.90
3.	Health Strategy	10	102.58
4.	Expert for health promotion in the county	6	16.65
5.	Scientist	6	62.07
6.	Chairperson senior advisory board	6	64.74
7.	Politician A	6	42.24
8.	Chairperson of an association	6	44.22
9.	Politician D	5	109.0
10.	Politician B	5	7.20
11.	Director regional sports department	5	32.5
12.	Director regional health department	4	2.0
13.	Working group	3	29.0
14.	Employee C municipality	2	29.0
15.	Politician E	2	29.0
16.	Employee A municipality	2	1.33
17.	GR+ office manager C	2	0.0
18.	Employee B municipality	2	0.0
19.	Employee D municipality	2	0.0

GR+ office manager C is a former member of the Healthy Region Plus.

**Table 3 ijerph-18-08372-t003:** Degrees and betweenness centrality as measured from the perspective of the steering committee; displayed are only actors with a betweenness centrality > 0; Density = 0.054).

	Actors/Artifacts	Degree	Betweenness Centrality
1.	Politician D	15	431.25
2.	Chairperson senior advisory board	11	383.54
3.	Politician A	10	429.21
4.	GR+ office manager A	8	276.0
5.	Chairperson of an association	4	113.25
6.	Employee D municipality	4	129.0
7.	Public relation	3	238.26
8.	Health conference	3	22.05
9.	Needs assessment	3	22.05
10.	Project B	3	118.0
11.	Politician H	3	162.0
12.	Expert for health promotion in the county	3	91.0
13.	Employee C municipality	3	93.0
14.	Project A	2	56.25
15.	Network B	2	22.05
16.	Health strategy	2	22.05
17.	Rules of procedure	2	22.05
18.	Politicians	2	105.0
19.	GR+ office manager B	2	164.0
20.	Project E	2	84.0

## Data Availability

The data presented in this study are available in depersonalized form on reasonable request from the corresponding author. The data are not publicly available due to confidentiality reasons and to allow for the participants’ anonymity.

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
