# Peer review of "Identification of Network Promoters in a Regional and Intersectoral Health Promotion Network: A Qualitative Social Network Analysis in Southern Germany"

_ijerph, 2021, doi:10.3390/ijerph18168372_

Round 1
Reviewer 1 Report
Thank you for the revision of the manuscript. The authors' research objectives and methods are now clear and understandable. On top of that, I have additional comment. In this study, two network mappings were conducted, but it is unclear (1)why the authors conducted two network mappings and (2)what the implications of the differences in the results are. Please specify these points in the methods and discussion section.
Reviewer 2 Report
This paper focus on an interesting topic of "intersectoral cooperation. In which, the author discussed members’ active participation in the network and obtained the conclusions that it is crucial to make network members advocates who promote the goals of a network in different roles, in diverse settings and across sectors.
Suggestion: the keywords of “intersectoral cooperation” should be shown in the title of the paper.
Author Response
Dear reviewer,
thank you very much for your review and the suggestion regarding the keyword in the title. We decided to keep the title but we changed the keyword "intersectoral action" to "intersectoral cooperation" instead.
Thank you!
Reviewer 3 Report
Social network analysis is not my area of expertise so I have little to say on the analysis itself,
but in summary, whilst the topic is of potential relevance (health promoting networks), there are a number of major issues with the paper. In particular, the description of methods is extremely think so it is not clear what has been done and as a result, it is unclear where the findings have come from and whether they hold any validity. In addition, there is no applicability of the findings to intersectoral health promotion, so the reader is left wondering what relevance they might have to practice. If the paper were to be considered for publication, these aspects would have to be addressed very clearly.
The introduction mentions importance of involving communities in shaping ‘health-in-all-policies approach’ (p.1 lines 31-32), yet no explanation offered as to why participatory methods then focus solely on institutional actors.
‘The objective of this exploratory study is to identify different types of network promoters that foster cooperation in intersectoral networks for health promotion’ – to what end? There is also no explanation as to why the Healthy Region Plus steering committee is chosen as the unit of analysis.
There are valuable points raised regarding the importance of acknowledging and identifying informalised network connections and interactions which contribute to the sustained integration of collaborative networks, for example how these can have a greater influence on work-relevant contexts than formal relationships (line 72). However the nature of these relationships are not clearly defined which makes it difficult to find the paper’s specific theoretical or applied contribution to health promotion. The promoter model underpinning the analytical framework should be situated within the context of health promotion to clarify its relevance to the field. This would help the reader understand what contribution is made by conceptualising these actors through the lens of power, expert, process and relationship promotion.
Methods
The epistemological approach should be discussed in the methods but it isn’t mentioned.
Before stating that you are using such an approach, you need to explain to the reader why you chose this approach. Is it the only one available? Is it well used? In what contexts? What evidence is there of its validity as a tool?
“Data collection was based on the net map approach” – basically, the only way to find out what has been done is to refer to the reference you have provided (behind a paywall). Even then, the reader would have to make a guess on what aspects of the referenced paper had been used. ’ I am English and I had to Google the term ‘stone checker’ but even then I couldn’t make sense of the sentence in terms of describing a recognized methodology Further searching around Net-mapping to try and understand this approach and how it might differ to conventional power-mapping exercises, and particularly what ‘stonecheckers’ might refer to, found Schiffer’s blog on the topic, where they describe a step in the process where participants build and assign ‘influence towers’ to actors in the network. I can only assume the stone checkers refer to the objects used to stack these towers? Remember that the methods section should allow the reader to go out and repeat your study in another context so it needs to provide enough detail to allow them to do so.
The rest of the methods section is a description of what was done (process followed) as opposed to describing the methodology. It is not clear from the methods that participants were asked to fill in self-evaluations prior to the group mapping exercise, this only becomes apparent in the results section where differences between linkages described in self-evaluations and perceived networks are discussed (lines 255-258).
Results and discussion
A table of the actors of the steering committee who participated in the network mapping is presented, but details of other kinds of and numbers of actors who make up the whole steering committee network but who did not participate in the data collection exercise is missing. The importance of missing actors is alluded to in the discussion (line 362), but not addressed in a meaningful way. Given that Social Network Analyses are very sensitive to missing data/nodes which can have significant effects on the network structures captured, it’s important to describe how missing actors and their attributes may have produced a very different looking network and what implications this may have on interpretation of results.
The mapping exercises describe asking participants to indicate ‘interactions’ between actors and their own ‘contributions’, which seem to be captured by writing down various ‘collaborations’ and ‘tasks’ on cards, but it’s not clear to me what relationship/ties are being captured and analysed here. For example, on line 248 they appear to describe ‘noteworthy impact of the network activities’ - how was ‘impact’ defined by the group, was this measure weighted when drawing connections? Were these disputed during mapping exercises and how was consensus reached? Without clear definitions of what these connections are representing, it’s not possible to determine what kind of network is being described and analysed.
What methodological and theoretical reason is there for conducting mapping exercises separately with different actors. E.g. a first network-mapping with the two/three administrative office managers and a second with steering committee members? One aim of the paper is to explore differences in these two perceived networks from the perspective of administrators and steering committee members, but I’m not sure this is justified and explained clearly enough in the methods.
There should also be some more discussion around the limitations of collecting network data in group exercises, particularly on topics of power and influence, as this can effect how participants respond. Differences in power dynamics or qualitative data exploring participant’s responses and experience of the exercise don’t appear to have been described or addressed. It feels problematic to have had administrators present but not participating in the second mapping exercise for example.
Some findings appear to be stating the obvious. That the administrators have most direct contacts and play a connecting role in terms of communication is a function of an administrative job description, but it doesn’t give the reader any sense of how the power dynamics work in the network, or what aspects of the process of the network might work well or not so well.
Given the authors write about the importance of incorporating qualitative methods for SNA, and which is presented in the discussion as a strength of the study to “gain in-depth information of meanings and processes in the respondents’ everyday life” (line 375), it was disappointing that no qualitative data was presented to contextualize the network structures captured such as the centrality and betweeness measures presented. It is stated that statistic results were contextualized by findings of qualitative interviews (line 176), but I see no discussion of how themes were coded and the findings from this thematic analysis. It is stated that “the question how relevant personal experiences and the motivation of the participating actors are for a successful network collaboration on the level of a steering committee” cannot be answered (lines 354-355), but this could have been a valuable contribution from the qualitative component had it been explored. The reader is left wondering what contribution the qualitative component of the research has made to the analysis.
It seems unusual and problematic to present actors and artefacts (events, products, services, projects) within the same visualization as the connections between them are presumably not comparable. The ties between each node isn’t defined in these visualisations, so it’s difficult to discern what kind of network we are actually looking at here.
The ‘driver-function’ is presented as a theme emerging from the qualitative analysis. Did this term arise from the transcripts, or is it a measure described in the literature on the promoter model? It hints at the importance of having a ‘relatively wide’ network (‘wide’ meaning number of/diversity of ties?) for information diffusion throughout the network. Quotes to illustrate this interpretation from the participants would be helpful.
The discussion section is quite abstract. It would be useful to bring in some examples of the promoter roles in practice, and identify how these roles might be applied in a health promoting network.
You also mention that your findings suggest that housing, mobility, environment, and spatial planning are potentially relevant but missing. What in your findings suggest this? Is this the view of the interviewees? If you are presenting qualitative data, you will need to also provide quotes etc. If you are able to explain how you came to this conclusion, it would be a useful finding for others working in the same field. Also, it is important not to introduce new findings in the discussion. Move this to your results section.
There is some confusion about the epistemological approach. You imply at the beginning of the study that an integrated approach will be used (p2 line 8) but there is no discussion of how you will integrate and the implications of this for your epistemological approach. This culminates in confusion in your discussion when you talk about control of bias followed by a statement that you are not looking for generalisability. This needs to be resolved.
Finally you state that “Our findings show that providing opportunity to contribute at every stage of network progress is crucial” but there is no evidence anywhere to suggest you have looked at the chronology of network building and neither do you discuss it in your results so again this comes as somewhat of a surprise. This would need to be covered in the results if it is a finding from the data and you would need to also explain in the methods how you have evaluated network progress.
Round 2
Reviewer 3 Report
Just for clarification, the first comment should have read 'The description of the methods is really thin' (not think). This was a typo. However, the authors have now addressed this.
I do think the qualitative findings would definitely be strengthened by including quotes, but I appreciate the authors would prefer to do this in a subsequent paper. I would suggest the two papers are clearly linked in some way as this will give the reader significantly more insight into the interpretation of the results.
This manuscript is a resubmission of an earlier submission. The following is a list of the peer review reports and author responses from that submission.
Round 1
Reviewer 1 Report
In this manuscript, the authors analyzed intersectoral cooperation in the "Healthy Region Plus" in Southern Germany using network mapping. Network analysis in health promotion is relatively new and the perspective of the study is interesting. However, this is a case report, not a scientific article. For the following reasons, the validity of this paper cannot be judged and its publication in IJERPH is considered inappropriate.
First, the article lacks information necessary for the reproducibility of the study (mainly in the Materials and Methods section). In the first place, because we do not know the potential scope of the network and potential stakeholders in the "Healthy Region Plus", the meaning of the analysis results cannot be properly determined. In addition, what is the definition of "intersectoral" in this study object? What makes it inner or outer? What role are the actors such as two office managers expected to play in Healthy Region Plus? Basic information is needed for those who is not familiar with "Healthy Region Plus" enough to understand the significance of this analysis.
Secondly, this analysis is not intended to be generalized. Since there is no control case for the analysis, the significance of the results about "Healthy Region Plus" cannot be conveyed. What contribution did the results make to the health promotion policies and the findings of the network analysis in general? In the analysis or discussion, it is necessary to compare the results with similar cases or in previous studies based on comparable evidence, both qualitative and quantitative.
Finally, the description itself is also incomplete. For example, what is the meaning of the color type in Figure 1 and 2? What are the values of degree, density, and betweenness obtained from the network analyses? What is the procedure to identify four types of promoters? If the values of degree, density, or betweenness is different for each of the four types, that would be meaningful. In addition, lines 264-267 and 289-291 could show important information, but there is no description of the results to support them.
Reviewer 2 Report
The work presented is of enormous interest and relevance. It is worth paying attention to how to improve cooperation between different sectors.
Below are some suggestions for improving the work presented:
- It is not very clear what the objective of the work is. The authors could clarify it.
- Similarly, the authors should organize the methodology. 10 interviews were conducted? It is not clear. Could they specify in a table the main characteristics of the participants? Why were 10 chosen?
- The authors do not mention the ethical aspects of the work. This is something that needs to be included in the manuscript.